# Pose Prior Learner:
# Unsupervised Categorical Prior Learning for Pose Estimation

## Abstract

A prior represents a set of beliefs or assumptions about a system, aiding inference and decision-making. In this paper, we introduce the challenge of unsupervised categorical prior learning in pose estimation, where AI models learn a general pose prior for an object category from images in a self-supervised manner. Although priors are effective in estimating pose, acquiring them can be difficult. We propose a novel method, named Pose Prior Learner (PPL), to learn a general pose prior for any object category. PPL uses a hierarchical memory to store compositional parts of prototypical poses, from which we distill a general pose prior. This prior improves pose estimation accuracy through template transformation and image reconstruction. PPL learns meaningful pose priors without any additional human annotations or interventions, outperforming competitive baselines on both human and animal pose estimation datasets. Notably, our experimental results reveal the effectiveness of PPL using learned prototypical poses for pose estimation on occluded images. Through iterative inference, PPL leverages the pose prior to refine estimated poses, regressing them to any prototypical poses stored in memory. Our code, model, and data will be publicly available.

## 1. Introduction

Priors represent beliefs or assumptions about a system or the characteristics of a concept. They are widely used in statistical inference (Lindley, 1961), cognitive science (Schad et al., 2021), and machine learning (Diligenti et al., 2017; Gülçehre & Bengio, 2016). This pre-existing

knowledge is essential for guiding the inference process, enabling AI models to make robust predictions in uncertain or ambiguous situations (Thiruvenkadam et al., 2008; Sung et al., 2015; Liang et al., 2024). The objective of our work is to enhance our understanding of priors in AI models and offer preliminary answers to the three key intelligence questions: (1) How do we acquire priors in the first place? (2) Can we learn them from input data in a self-supervised manner? (3) Can we enhance the quality of the priors? To tackle these questions, we first introduce the challenge of unsupervised categorical prior learning in the context of pose estimation from images. See **Figure 1** for the schematic illustration of the challenge. Categorical pose estimation is a classical computer vision task that identifies the structure of objects belonging to the same category by detecting their keypoints. A pose prior summarizes the common characteristics shared by a variety of poses. It encapsulates the expectation of the keypoint configurations and the connectivity between keypoints.

In parallel to our challenge of unsupervised categorical prior learning from images for pose estimation, unsupervised pose estimation leverages the abundant, unannotated visual information available in large image datasets to extract pose information (Hu & Ahuja, 2021; Sommer et al., 2024; Chen et al., 2019; He et al., 2022a; Schmidtke et al., 2021). The use of pose priors can provide valuable guidance in this process. We categorize the existing works in unsupervised pose estimation into two groups: those that incorporate hand-made priors and those that operate without any priors.

Recent approaches (He et al., 2022a; Sun et al., 2022; 2023) attempt to predict keypoints from images, construct object structure representations using these keypoints, and learn effective structural information through image reconstruction. However, without pose priors, these methods can be disrupted by background information or may predict infeasible topological configurations of an object during occlusion. The risk of generating inaccurate keypoints stems from the absence of supplementary information that could help refine both keypoint localization and the connections between them.

The other group of methods (Schmidtke et al., 2021; Yoo & Russakovsky, 2023) utilize prior knowledge of

---

[1]Anonymous Institution, Anonymous City, Anonymous Region, Anonymous Country. Correspondence to: Anonymous Author <anon.email@domain.com>.

Preliminary work. Under review by the International Conference on Machine Learning (ICML). Do not distribute.

*Figure 1.* **Schematic illustration of the unsupervised categorical prior learning challenge in the task of pose estimation.** Given a series of images (framed in blue), the challenge is to learn the pose prior (green circle) in a self-supervised manner. The pose prior comprises keypoint and connectivity priors. To address this challenge, we propose a Pose Prior Learner (PPL). During PPL training, prototypical poses in the memory (blue circle) are aggregated from the individual poses estimated from all images (orange circle) and distilled into a general pose prior. The pose prior can then guide pose estimation through transformations. This cyclic process strengthens the learning of robust pose prior representations, resulting in more accurate pose estimations, which, in turn, helps capture more representative prototypical poses. During inference, the learned pose prior refines pose estimation on occluded images (bounded in red), regressing them to the prototypical poses stored in memory. Blue arrows illustrate signal flows during PPL training, while red arrows indicate signal flows during inference on occluded images.

a category's general pose to guide the pose estimation of individuals within that category. Conceptually, each category is expected to exhibit a generalized and distinctive pose prior that reflects characteristics such as shape, size, and structure. Individual poses should be seen as geometric transformations of this category-specific pose prior. As a result, employing a category-specific pose prior aids in guiding and regularizing the learning of poses. However, obtaining comprehensive general pose priors is highly challenging, as it requires extensive human annotations, particularly for novel categories. Moreover, human annotations may introduce implicit biases, hindering models from learning more meaningful priors.

Loosely inspired by how humans develop a general prior representation of an object category by observing individual object instances in images and subsequently using them to infer upcoming individual poses, we propose a new method called the Pose Prior Learner (PPL). PPL is designed to effectively learn a meaningful pose prior for a certain object category. It utilizes a hierarchical memory to store a finite set of prototypical poses and extract a general pose prior from them. Initially, both the hierarchical memory and the prior are randomly initialized but learnable parameters. During training, effective pose learning is supervised through image reconstruction. As training progresses, the hierarchical memory retains and aggregates multiple accurate prototypical poses, thereby contributing to a more precise pose prior and enhancing the model's ability to estimate poses.

Upon completing the training, we obtain a model that enables accurate pose estimation, a categorical pose prior

that encapsulates the general features of a category, and a hierarchical memory that stores diverse prototypical poses for that category. We evaluate the effectiveness of our PPL across several human and animal pose estimation benchmarks. We visualize their pose priors to further interpret what our approach has learned. Additionally, we introduce an iterative inference strategy to estimate the poses of objects in occluded scenes using the trained hierarchical memory and the pose prior. Our contributions are highlighted below:

**1.** We introduce the challenge of unsupervised categorical prior learning in the context of pose estimation.

**2.** We propose a new method called Pose Prior Learner (PPL) for unsupervised pose estimation. PPL outperforms existing methods across several pose estimation benchmarks and offers explainable visualizations of pose priors. Notably, We found that predefined human priors are not always optimal. Our PPL even outperforms models using human-defined priors.

**3.** During inference, we utilize an iterative strategy in which PPL progressively leverages priors to refine estimated poses by regressing them to the nearest prototypical poses stored in memory. Experimental results demonstrate that our PPL accurately estimates poses, even in occluded scenes.

## 2. Related Works

**Unsupervised Pose Estimation without Priors.** Numerous unsupervised learning methods without priors have been proposed to detect keypoints from images, which are then used to reconstruct images for supervision (Li et al., 2021;

Geng et al., 2021; Zhang et al., 2018; Sun et al., 2022; Thewlis et al., 2017; Jakab et al., 2020). For example, AutoLink (He et al., 2022a) extracts keypoints from the image and estimates the strength of the links between pairs of keypoints. It then combines these keypoints with the link heatmap to reconstruct the randomly masked image. In these methods, keypoints are directly predicted from the image and supervised solely by image reconstruction, leading to potential detection of keypoints in background regions with complex textures. To alleviate this problem, BKind (Sun et al., 2022; 2023) uses keypoints extracted from two video frames to reconstruct the pixel-level differences between these two frames. However, the lack of constraints on keypoint configuration and connectivity still undermines the reliability of their approach. In contrast, our PPL utilizes the learned pose prior as a constraint to mitigate these issues.

**Unsupervised Pose Estimation Incorporating Human-defined Priors.** Several methods utilize prior knowledge from human annotators to guide the pose estimation (Chen & Dou, 2021; Shi et al., 2023; Zhang et al., 2022; Schmidtke et al., 2021; Yoo & Russakovsky, 2023). Among these methods, Shape Template Transforming (STT) (Schmidtke et al., 2021) applies affine transformations to a predefined pose prior, aligning it with the estimated pose from a video frame. By incorporating an additional frame from the same video to provide background information, an image reconstruction loss supervises the pose estimation process. The pose prior effectively guides pose estimation by constraining the shape of the human pose and the connectivity between body parts. However, pose priors are often difficult to obtain, requiring costly human annotations. Moreover, HPE (Yoo & Russakovsky, 2023) has shown that predefined pose priors are not always optimal, and tuning the shape of the prior can sometimes improve performance. Unlike these methods, our approach learns the prior directly from input images without any manual annotations, and models with our learned priors even outperform those using human-defined priors.

**Compositional Memory Architectures.** Compositional memory has been widely used in many computer vision tasks, such as question answering (Seong et al., 2021), object segmentation (Seong et al., 2021), and sence graph generation (Deng et al., 2022). In pose estimation, PCT (Geng et al., 2023) decomposes a human pose into discrete tokens, where each token connects several interdependent joints and characterizes a sub-structure of the entire human pose. This approach is highly effective for decomposing and reconstructing poses, providing robust pose representations while significantly reducing computation and storage costs. However, PCT encodes all tokens into the same embedding space, making it difficult to aggregate semantic tokens that represent different sub-structures. In contrast, our PPL employs a compositional hierarchical memory, which parses poses into memory banks. Each memory bank explicitly contains multiple vectors encoding the variations of each sub-structure of a pose. This facilitates the aggregation of poses into a representative categorical pose prior.

## 3. Our Proposed Pose Prior Learner (PPL)

We introduce our proposed method, Pose Prior Learner (PPL). Given images featuring object instances from a specific category, such as dogs or humans, PPL can accurately estimate the poses of the objects in that category while gradually learning a general pose prior through unsupervised learning. Note that our PPL requires no extra knowledge from human annotators. The architecture of PPL is presented in **Figure 2**.

Mathematically, we represent the topology for an object as a graph connecting keypoints with shared link weights, also known as connectivity. For a category of objects, its general pose prior $V$ is defined as $V = (T, W)$, where $T$ represents its keypoint prior and $W$ denotes the connectivity prior. Specifically, $T$ consists of $N$ keypoints: $T = [P_1, P_2, ..., P_N]$, where $P_i \in [-1, 1] \times [-1, 1]$ is the normalized 2D coordinates in the image pixel space. Unlike pre-defined priors, before training, $P_i$ does not explicitly encode the semantic parts of an object. $W$ is a $2D$ matrix of size $N \times N$, where each entry $w_{ij}$ in the matrix represents the connectivity probability between two keypoints $P_i$ and $P_j$. For instance, in the case of humans, the hand is connected to the torso via an arm; thus, the connectivity probability between these two parts should be higher than the connectivity probability between a hand and a foot. We initialize $W$ with random positive values. $T$ is decoded from a hierarchical memory $M$ storing compositional parts of prototypical poses, which is also randomly initialized.

During training, PPL inputs an image $I$ of $H \times W \times 3$ where H and W are the height and the width. The aim of PPL is to learn to correctly predict the image-specific keypoints $T'$ and their connectivity on $I$ from the general categorical pose prior $V = (T, W)$. Ideally, if PPL makes perfect predictions of the pose on $I$, by combining it with the background information, the reconstructed image $I_{\text{recon}}$ should match $I$ exactly. The background information is provided by the reference image $I_{ref}$. For images extracted from video datasets, $I_{ref}$ can be a randomly selected frame from the same video that features the same object in a different pose. Alternatively, for a static image dataset, $I_{ref}$ can be a randomly masked version of the original image. Next, we introduce how we estimate $T'$ on $I$ using the keypoint prior $T$.

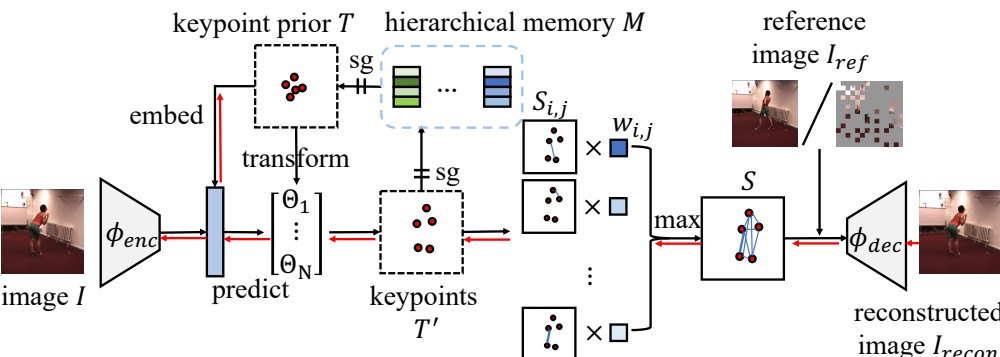

*Figure 2.* **Overview of our proposed Pose Prior Learner (PPL).** We first distill the keypoint prior from the hierarchical memory $M$. Features of the image $I$ and the embedding of the keypoint prior are concatenated to predict the affine transformation parameters. The keypoint prior is transformed and their pair-wise links are modulated with the connectivity prior $W$ to obtain the combined link heatmap $S$. The concatenation of the link heatmap $S$ and the reference image $I_{ref}$ is decoded to produce the reconstructed image $I_{recon}$. The $sg$ symbol represents the stopping gradient operation. The red arrows indicate the gradient flows during backpropagation based on image reconstruction. See **Section 3.4** for training details.

### 3.1. Transformation of the Keypoint Prior

Given the image $I$, we use a feature extractor $\phi_{enc}$ to extract its embedding $h_I$: $h_I = \phi_{enc}(I)$. $\phi_{enc}$ is a 2D-Convolution Neural Network (2D-CNN) trained from scratch. The keypoint prior $T$ is converted into an embedding $h_T$ via a series of fully connected layers. Together with $h_T$ as inputs, PPL learns to predict the affine transformation parameters $\Theta_i \in [\Theta_1, \Theta_2, ..., \Theta_N]$ for each keypoint $P_i$ in $T$ from $h_I$ via a two-layer fully connected network denoted as $FC(\cdot)$:

$$[\Theta_1, \Theta_2, ..., \Theta_i, ..., \Theta_N] = FC(h_I, h_T),$$

$$\text{where } \Theta_i = \begin{bmatrix} a^{(i)} & b^{(i)} & t_x^{(i)} \\ c^{(i)} & d^{(i)} & t_y^{(i)} \\ 0 & 0 & 1 \end{bmatrix}. \quad (1)$$

$t_x^{(i)}$ and $t_y^{(i)}$ are the translations and $a^{(i)}, b^{(i)}, c^{(i)}, d^{(i)}$ are the coefficients that define rotation, scaling, and shear. Each point $P_i$ in $T$ can then be transformed by $\Theta_i$, resulting in the image-specific keypoints $T'$ for the image $I$:

$$T' = [P'_1, P'_2, ..., P'_N], \text{ where } [P'_i, 1]^\top = \Theta_i[P_i, 1]^\top. \quad (2)$$

### 3.2. Connecting Keypoints Based on the Connectivity Prior

The connectivity of keypoints in objects is often fixed and rigid, for example, human arms maintain a relatively constant length, with a hand always connected to the torso via an arm and never connected to a foot. This rigidity in connectivity serves as a constraint, aiding in the regularization of pose estimation. In this section, we introduce the connectivity prior and explain how it can be used to regularize the connectivity strength between any pair of estimated keypoints in $T'$ on $I$.

Similar to AutoLink (He et al., 2022a), PPL connects any two keypoints $P'_i$ and $P'_j$ in $T'$ to obtain differentiable link heatmap $S_{i,j} \in \mathbb{R}^{H \times W}$. Intuitively, each 2D link heatmap represents a probability density map, where the pixel values along the link between two points are high, while other areas are assigned values close to zero. For any point $P'_i$, its strongest connectivity to any of the other points in $T'$ is activated on the combined link heatmap $S \in \mathbb{R}^{H \times W}$ via a max pooling operation over all the $N \times N$ link heatmaps:

$$S = \max_{i,j}^{N \times N}(w_{i,j} S_{i,j}), \quad (3)$$

where $w_{i,j}$ in the connectivity prior $W$ modulates the link heatmap $S_{i,j}$ based on whether the two keypoints $P'_i$ and $P'_j$ are physically connected. Ideally, if PPL correctly estimates the probability of physical links for an object category, $S_{i,j}$ will receive higher connectivity values, thereby activating the locations linking these two keypoints on the combined link map $S$.

Given the combined link map $S$ and the reference image $I_{ref}$, PPL can reconstruct the image $I$. $I_{ref}$ provides background information for reconstruction, while $S$ supplies foreground structural information by linking all the estimated keypoints with the connectivity prior. Therefore, we concatenate $S$ and $I_{ref}$ and feed them into a 2D-CNN to perform the image reconstruction $I_{recon}$, where $I_{recon} = \phi_{dec}(I_{ref}, S)$.

### 3.3. Reconstruction of Keypoint Configuration with Memory

$M$ is a memory module storing compositional representations of prototypical poses. $M$ is hierarchical because it consists of $m$ memory banks $\{b_1, b_2, ..., b_m\}$, each containing $k$ learnable vectors of dimension $d$. This setup allows for efficient retrieval and management of information at multiple levels, where $M$ can represent pose configurations while memory banks capture specific pose

sub-structures. This hierarchy enables $M$ to effectively learn robust keypoint prior representations $T$ for the following reasons: (1) By storing multiple prototypical poses, memory helps in aggregating these poses, which aids in creating a more robust and comprehensive prior that captures variations within an object category. (2) $M$ can assist in organizing information hierarchically, making it easier for PPL to retrieve relevant prototypical poses when making predictions in uncertain or ambiguous scenarios, such as occlusion. and (3) By leveraging $M$, PPL can refine its predictions iteratively, using stored poses to adjust its outputs based on previously learned pose representations. Next, we introduce how $M$ is structured.

Given the estimated $N$ keypoints in $T'$, we first encode them into several tokens of dimension $d$ with several MLP-Mixer blocks (Tolstikhin et al., 2021), denoted as $MIX_{enc}$. The number of encoded tokens should always be equal to the number of memory banks, as we design these tokens to have different embedding space. Each token $g_i$ corresponds to the embedding of memory bank $b_i$ in memory $M$. We define $G$ as the collection of the $m$ tokens:

$$G = [g_1, g_2, ..., g_m] = MIX_{enc}(T'), \qquad (4)$$

If all the vectors in all the memory banks of $M$ learn to capture unique parts of prototypical poses, each $g_i$ should be able to retrieve the most similar vector in the memory bank $b_i$ of $M$ and reconstruct $G$ itself. Here, we use $L2$ distance to measure the similarity between $g_i$ and $k$ vectors in $b_i$ of $M$. The vector in $b_i$ that is most similar to $g_i$ is denoted as $g_i'$. Thus, for each $g_i$, we can always find the corresponding $g_i'$ from $b_i$, resulting in a collection of $G' = [g_1', g_2', ..., g_m']$. Another series of MLP-Mixer blocks (denoted as $MIX_{dec}$) is then used to decode $G'$ back to $N$ keypoints $T'_{recon}$, where $T'_{recon} = MIX_{dec}(G')$. See **Figure A4(a)** in **Appendix A5** for the schematic.

Unlike PCT (Geng et al., 2023) where all vectors of a memory are within the same embedding space, We organize $M$ into $m$ memory banks, each containing $k$ vectors that represent an independent embedding space. This structure enables us to efficiently distill each memory bank to form a general pose representation, which can be viewed as the keypoint prior $T$. Specifically, we define $MP(\cdot)$ as the mean pooling operation. PPL pools the $k$ vectors of each memory bank into one vector. These pooled $m$ vectors are further decoded by $MIX_{dec}$ into $N$ points of our keypoint prior $T$. See **Figure A4(b)** in **Appendix A5** for the schematic of distilling $T$: $T = MIX_{dec}([MP(b_1), MP(b_2), ..., MP(b_m)])$.

Compared to simply using a moving average of the learned poses, the advantages of employing hierarchical memory and keypoint configuration reconstruction are as follows: (1) the hierarchical memory increases the complexity of the

embedding space with compositional parts, enhances the expressive capability for poses and facilitates more accurate pose retrieval for occluded images; (2) the compositional nature of the hierarchical memory addresses data bias, enabling the distillation of a less biased prior.

### 3.4. Training and Inference

Our PPL is trained to jointly minimize all the four losses: the image reconstruction loss $L_{ir}$, the boundary loss $L_b$, the link regularization loss $L_l$, and the keypoint configuration reconstruction loss $L_{kr}$. We elaborate on these four losses below. Additionally, we use three training techniques for stable convergence of the network. We present the details of these training techniques in **Appendix A6**.

**Image Reconstruction Loss.** If PPL correctly estimates the pose on the original image $I$, the reconstructed image $I_{recon}$, based on the estimated pose, should be identical to $I$. Therefore, ensuring the quality of $I_{recon}$ encourages PPL to improve its pose estimation accuracy. To achieve this, we apply a perceptual loss on the embeddings of $I$ and $I_{recon}$, extracted using a frozen feature extractor $\psi(\cdot)$ from the VGG19 network pre-trained on ImageNet (Russakovsky et al., 2015). The perceptual loss is defined as: $L_{ir} = \|\psi(I_{recon}) - \psi(I)\|_1$.

**Boundary Loss.** To ensure that the network does not transform the points in the keypoint prior outside the boundaries of the image, we limit the x and y coordinates of the transformed keypoints to be within the image:

$$L_b = \sum_{* \in x, y} \begin{cases} |P'_{i,*}| & \text{if } |P'_{i,*}| > 1, \\ 0 & \text{otherwise.} \end{cases} \qquad (5)$$

where $P'_{i,x}$ and $P'_{i,y}$ are the normalized $x$ and $y$ coordinate of the keypoint $P'_i$ respectively.

**Link Regularization Loss.** A person's arm always maintains a fixed length regardless of the poses. Thus, we propose the constraint that links should be assigned a high weight if they do not vary significantly in length before and after the affine transformation. The loss $L_l$ encourages the preservation of the link lengths during pose estimation. It is defined as in the equation below, where $l(\cdot)$ is the $L2$ distance between two keypoints before and after the affine transformation:

$$L_l = \sum_{i,j} w_{i,j} \|l(P_i, P_j) - l(P'_i, P'_j)\|_1. \qquad (6)$$

**Reconstruction Loss on Keypoint Configurations.** In **Section 3.3**, given a collection of token representations in $G$, PPL retrieves the most similar vectors from each memory bank of the hierarchical memory $M$ and generates $G'$ in a non-differentiable manner. To ensure that $M$ learns to store

*Table 1.* **Keypoint detection on CUB-200-2011.** We report the mean $L2$ error normalized by the image resolutions of $128 \times 128$. We use 10 keypoints for CUB-Aligned and 4 keypoints for CUB-001, CUB-002, CUB-003, and CUB-all. The best is in bold.

| Method | CUB-aligned | CUB-001 | CUB-002 | CUB-003 | CUB-all |
|---|---|---|---|---|---|
| (Zhang et al., 2018) | 5.36 | 26.9 | 27.6 | 27.1 | 22.4 |
| (He et al., 2021) | 5.21 | 22.6 | 29.1 | 21.2 | 14.7 |
| (He et al., 2022b) | 3.23 | 22.1 | 22.3 | 21.5 | 12.1 |
| AutoLink (He et al., 2022a) | 3.51 | 20.2 | 19.2 | 18.5 | 11.3 |
| PPL (ours) | **3.19** | **19.3** | **18.6** | **17.3** | **10.5** |

*Table 2.* **Keypoint detection on Human3.6m.** We report the mean $L2$ error normalized by the image resolutions (Res.) of both $256 \times 256$ and $128 \times 128$. ∗ is the result trained by using masked images as reference images. The best is in bold.

| Method | Res. | Norm. $L2$ Error |
|---|---|---|
| (Jakab et al., 2020) | 256 | 2.73 |
| (Thewlis et al., 2017) | 256 | 7.51 |
| (Lorenz et al., 2019) | 256 | 2.79 |
| (Zhang et al., 2018) | 256 | 4.91 |
| (Schmidtke et al., 2021) | 256 | 3.31 |
| AutoLink (He et al., 2022a) | 128 | 2.76 |
| PPL (ours) | 128 | **1.92** |
| PPL* (ours) | 128 | **2.23** |
| PPL (ours) | 256 | **2.56** |

meaningful token embeddings that represent compositional parts of poses, the retrieved vectors $G'$ from $M$ should closely match $G$. Moreover, if these compositional parts are structured correctly, the vectors should be able to decode into meaningful keypoint configurations $T'_{recon}$ that are close to the original keypoint configurations $T'$. Therefore, we introduce the keypoint configuration reconstruction loss defined as: $L_{kr} = \|T'_{recon} - T'\|_2 + \|G - G'\|_2$.

*Table 3.* **Keypoint detection on Taichi.** For consistency with the baseline methods, we report the summed $L2$ error at a resolution of $256 \times 256$. ∗ is the result trained by using masked images as reference images. The best is in bold.

| Method | Summed $L2$ Error |
|---|---|
| (Siarohin et al., 2021) | 389.78 |
| (He et al., 2021) | 437.69 |
| (Zhang et al., 2018) | 343.67 |
| (He et al., 2022b) | 417.17 |
| AutoLink (He et al., 2022a) | 316.10 |
| PPL* (ours) | **298.60** |
| PPL (ours) | **293.35** |

**Iterative Inference.** We propose an iterative inference strategy (**Figure 3**). 4 iterations are used for every experiment. In every iteration, we take the reconstructed image $I_{recon}$ from the last iteration (the original image $I$ for iteration 0) as the input. We infer its keypoints $T'$ as the output keypoints of the current iteration. The hierarchical memory $M$ is used to reconstruct $T'$ and the reconstructed keypoints $T'_{recon}$ are used to obtain the reconstructed image

$I_{recon}$. $I_{recon}$ is then used as the input for the next iteration. We keep the original occluded image $I$ as the reference image for all the iterations. See **Appendix A1** for more implementation details.

## 4. Experiments

For quantitative experiments, we use three image datasets: Human3.6m (Ionescu et al., 2013), Taichi (Siarohin et al., 2019b;a), and CUB-200-2011 (Wah et al., 2011). For fair comparisons, we set the number of keypoints $N$ to be the same as (He et al., 2022a). For qualitative visualizations, we use Youtube dog videos, Flowers (Nilsback & Zisserman, 2008), 11k-Hands (Afifi, 2019), and Horses (Zhu et al., 2017). See **Appendix A2** for details. We utilize video frames as $I_{ref}$ for the Human3.6m, Taichi, and YouTube dog videos, while randomly masked images are used as $I_{ref}$ for other datasets. On Human3.6m and CUB-200-2011, we report the results in the mean $L2$ error between the predicted keypoints and the ground truth, normalized by the image size. For Taichi (He et al., 2022a; Siarohin et al., 2021; He et al., 2021; 2022b; Zhang et al., 2018), we use the summed $L2$ error computed at a resolution of $256 \times 256$.

### 4.1. Unsupervised Categorical Pose Estimation

We compare the keypoint detection results of PPL with other unsupervised pose estimation methods and present the results in **Table 1** (CUB-200-2011), **Table 2** (Human3.6m), and **Table 3** (Taichi). On all datasets, PPL significantly outperforms all baselines across all image resolutions. Among the baselines, AutoLink (He et al., 2022a) also incorporates learnable connectivity priors. However, its performance is inferior due to the absence of hierarchical memory. Notably, even when masked images are used as $I_{ref}$, PPL consistently outperforms AutoLink. This demonstrates that the effectiveness of PPL is primarily attributed to its pose prior, rather than relying on cross-frame reconstruction. Additionally, we note that STT (Schmidtke et al., 2021), a baseline with human-defined priors, still underperforms PPL. Consistent with HPE (Yoo & Russakovsky, 2023), this suggests that pre-defined priors are not always optimal. PPL can learn more representative priors that outperform those manually defined.

**Visualization of Pose Estimation.** We provide the visualization results of pose estimation on Human3.6m (**Figure A5(a)**), YouTube dog videos (**Figure A5(b)**), 11k-Hands (**Figure A5(c)**), CUB-200-2011 (**Figure A5(d)**), Horse (**Figure A5(e)**), and Flowers (**Figure A5(f)**) in **Appendix A7**. The visualization results demonstrate that PPL can learn categorical priors and estimate poses for various object categories, without any external annotations. For example, in Row 2, Column 2 of **Figure A5(a)**, PPL correctly estimates the bowing pose of a person.

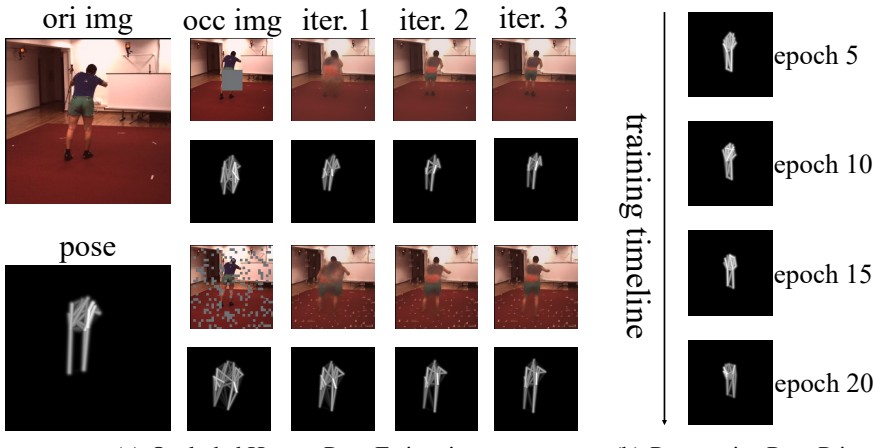

*Figure 3.* **Overview of the iterative inference strategy in our PPL (Section 3.4).** During inference, we iteratively use the reconstructed image $I_{recon}$ as input to estimate the pose $T'$. The hierarchical memory $M$ refines the estimated pose $T'$ and outputs $T'_{recon}$. With the original image $I$ as the reference image $I_{ref}$, PPL reconstructs the image $I_{recon}$. It is then used as the input image in the next iteration.

(a) Occluded Human Pose Estimation.   (b) Progressive Pose Priors.

*Figure 4.* **Visualization results of poses estimation on occluded images and progressive pose priors of PPL on Human3.6m.** (a) Pose estimation on occluded images in Human3.6m. The first column shows the original image and its estimated pose by PPL. Columns 2-5 show the iterative inference process where the reconstructed images by PPL (Row 1 and 3) are fed back to itself for estimating poses (Rows 2 and 4) on occluded images either using CenterMasking (Row 1 and 2) or RandomMasking (Row 3 and 4). (b) The pose prior evolves as a function of training epochs (from top to bottom).

In Row 2, Column 5 of **Figure A5(b)**, PPL correctly estimates the pose of a dog lowering down its head. Moreover, we found the quality of estimated poses of dogs is inferior to that of humans. This is primarily due to the significant morphological differences among various dog breeds. Additionally, dogs often perform actions such as turning around, which can lead to significant changes in pose that are difficult to accurately capture by our priors in 2D space. Additional visualization results for Flowers, Hands, and Horses in **Appendix A7** further demonstrate that our PPL can be applied for various categories. Notably, flowers are rigid objects with no degrees of freedom like animals. Yet, our PPL still learns meaningful categorical priors for them.

**Visualization of the Pose Prior Changing with the Training Epochs.** We visualize the progressively learnt pose priors by our PPL as a function of training epochs. **Figure 4(b)** illustrates that the keypoint prior converges to a human shape by the early stage of training (epoch 5). Notably, the learnable keypoints align with the human joints defined in the literature, and the connectivity among keypoints corresponds to the physical connections between body parts. As training continues, the connectivity prior gradually learns the skeletal structure of the human body, with irrelevant links between keypoints diminishing over time, as seen when comparing epochs 15 and 20.

### 4.2. Ablation Studies

**Ablation on Prior Variants.** Here, we investigate how different initializations of connectivity and keypoint priors affect pose estimation and assess whether further refining these priors enhances performance. From **Table 4**, we obtain several key insights: (1) Models with frozen, human-defined priors (Column 4) perform worse than our PPL, indicating that PPL learns more representative priors than those predefined by humans. (2) Refining pre-defined keypoint and connectivity priors (Column 1) outperforms our default PPL, suggesting that PPL can enhance models with human-defined priors through refinement. (3) Interestingly, randomly initializing either keypoint or connectivity priors, followed by refinement during

*Table 4.* **Keypoint detection results of our PPL variants on the Human3.6m dataset.** All results in mean $L2$ errors are normalized by the image resolution of $256 \times 256$. Both keypoint prior (Row 1-2) and Connectivity prior (Row 3-4) can be either pre-defined (Pre.) or randomly initialized (Rand.). During training, the parameters in both the priors can be either frozen (✗) or learnable (✓). The last column (From Mem) shows the result of our default PPL method. Its keypoint prior is initialized from memory (From Mem). Its connectivity prior is randomly initialized (Rand.) and learnable (✓) during training. Best is in bold.

| | | 1 | 2 | 3 | 4 | 5 | 6 | 7 | 8 | 9 | 10 | 11 |
|---|---|---|---|---|---|---|---|---|---|---|---|---|
| Keypoint prior | Initialization | Pre | Pre | Pre | Pre | Pre | Pre | Rand | Rand | Rand | Rand | From mem |
| | Trainable | ✓ | ✓ | ✗ | ✗ | ✓ | ✗ | ✓ | ✗ | ✓ | ✗ | ✗ |
| Connectivity prior | Initialization | Pre | Pre | Pre | Pre | Rand | Rand | Pre | Pre | Rand | Rand | Rand |
| | Trainable | ✓ | ✗ | ✓ | ✗ | ✓ | ✓ | ✓ | ✓ | ✓ | ✓ | ✓ |
| Normalized $L2$ Error | | **2.51** | 2.66 | 2.58 | 2.70 | 2.54 | 2.61 | 2.68 | 2.72 | 2.75 | 2.83 | 2.56 |

training (Columns 5-9), yields comparable performance to models with human-defined priors. This suggests that human-defined priors may not be necessary for effective pose estimation. (4) Surprisingly, freezing randomly initialized keypoint priors also results in reasonable pose estimation accuracy, though it is still lower than PPL's default performance (Columns 7 and 9). (5) In contrast to (4), freezing random connectivity priors prevents the model from converging, implying that connectivity priors play a more critical role in guiding pose estimations than keypoint priors.

**Ablation on Number & Dimension of Vectors in Each Memory Bank.** In our hierarchical memory, we used 34 memory banks. Here, we analyze the impact of the number of vectors per memory bank and the dimension of each vector on PPL's pose estimation performance. From **Figure A1** in **Appendix A3**, we observed that PPL remains robust across different vector counts and dimensions, although performance slightly improves with more vectors of higher dimensions. As a result, we fixed 16 vectors per memory bank, each with a dimension of 512, for all experiments.

**Ablation on Number of Keypoints.** We varied the number of keypoints in the pose priors from 4 to 32. The results in **Figure A1** in **Appendix A3** show that pose estimation accuracy improves as the number of keypoints in the prior increases. However, using 32 keypoints offers limited improvement compared to PPL with 16 keypoints.

### 4.3. Pose Estimation in Occluded Scenes

To verify the robustness of PPL in occluded scenes, we divide the image into $32 \times 32$ patches and apply two masking techniques: RandomMasking and CenterMasking. In RandomMasking, we randomly mask a certain proportion of image patches, with the proportion ranging from 0.1 to 0.4. In CenterMasking, we mask only the center region of the image, gradually increasing the masking size from $4 \times 4$ to $12 \times 12$ patches. We explore the effect of occluded areas on PPL. From **Figure A2** and **Figure A3** in

**Appendix A4**, we observe that at iteration 0, as the occluded areas increase, overall performance declines with larger occlusions. However, with our iterative inference strategy, PPL effectively infers the missing parts of the poses by utilizing prototypical poses stored in hierarchical memory and the learned priors. Notably, it restores partially occluded poses to reasonably complete full-body poses, leading to a lower $L2$ error, comparable to those without occlusion. This effect is more pronounced with smaller occluded areas.

**Visualization of Pose Estimation with Occlusion.** We present the estimated poses by our PPL for occluded images as a function of the number of inference iterations in **Figure 4(a)**. Across both RandomMasking and CenterMasking, with our iterative inference strategy, PPL successfully reconstructs the occluded image parts after three iterations and meanwhile, predicts reasonable full-body poses.

## 5. Discussion

We introduce the challenge of unsupervised categorical prior learning and highlight its significance in pose estimation. To address this, we propose a novel method called Pose Prior Learner (PPL). PPL utilizes a hierarchical memory to store compositional parts of learnable prototypical poses, which are distilled into a general pose prior for any object category. Our experimental results show that PPL requires no additional human annotations and outperforms recent competitive baselines in pose estimation. Notably, the learned prior proves to be even more effective in pose estimation than methods that rely on human-defined priors. With hierarchical memory and learned priors, PPL can perform iterative inferences and robustly estimate poses in occluded scenes. Despite outstanding performance in unsupervised categorical pose estimation, PPL has several limitations. For instance, it learns 2D priors, which makes it difficult to capture real-world 3D postures. Thus, PPL struggles in scenarios where objects involve rotations or significant shape changes. Extending PPL to incorporate 3D priors will be a key focus of our future research.

## 6. Impact Statement

We present the challenge of unsupervised learning of categorical priors for pose estimation and propose Pose Prior Learner (PPL) to address it. By leveraging learned priors, PPL enhances robustness and generalizability during inference, promoting the safe deployment of AI technologies. However, privacy concerns surrounding images and videos in these datasets, especially for human pose estimation, must be carefully addressed. Protecting personal and sensitive information remains paramount as AI advances.

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

## A1. Implementation Details of our Pose Prior Learner

We use the Adam optimizer with a learning rate of $10^{-3}$ and a batch size of 64, training for 50 epochs. Unless specified, all images are resized to $256 \times 256$. The learning rate for link weights is scaled by 512 to address the small gradients of SoftPlus near zero. We conduct experiments using the link thickness $\sigma^2 = 5 \times 10^{-4}$ across all benchmark datasets, where we adopt the same definition of $\sigma$ used in (He et al., 2022a). For the hierarchical memory, we use 34 memory banks, each of which contains 16 vectors of dimension 512, for all experiments.

## A2. Datasets

**Human3.6m** (Ionescu et al., 2013) is a standard benchmark dataset for human pose estimation, consisting of 3.6 million video frames. These frames include both 3D and 2D keypoints and were captured in a controlled studio environment with a static background, featuring various actors. We adhere to the approach outlined in (Zhang et al., 2018; He et al., 2022a), focusing on six activities: direction, discussion, posing, waiting, greeting, and walking. For training, we use subjects 1, 5, 6, 7, 8, and 9, and for testing, we use subject 11.

**Taichi** (Siarohin et al., 2019b;a) consists of 3,049 training videos and 285 test videos featuring individuals performing Tai-Chi, with diverse foreground and background appearances. Following the approach in (Siarohin et al., 2021), we use 5,000 frames for training and 300 frames for testing.

**YouTube dog videos** are videos with green backgrounds collected from YouTube to further qualitatively demonstrate the performance of PPL. Existing dog datasets are not suitable, as they often contain multiple, partially occluded dog instances. Therefore, we curated a custom dataset using 20 YouTube videos, extracting 2,000 frames for training. This dataset allows us to demonstrate PPL's ability to learn pose priors for non-human categories without using ground truth poses. All images are trained and tested at a resolution of $256 \times 256$. We use 10 keypoints for this category and provide the visualization of the estimated poses on the test videos. The dataset will be publicly released together with other data, models, and source code.

**CUB-200-2011** (Wah et al., 2011) contains 11788 images of birds. We crop and align the images according to (He et al., 2022a). We use the train/val/test split of (Choudhury et al., 2021). All images are trained and tested at a resolution of $128 \times 128$.

**11k-Hands** (Afifi, 2019), **Horse** (Zhu et al., 2017), and **Flowers** (Nilsback & Zisserman, 2008) are used for qualitative visualization. Images with multiple horses are removed and all horses are aligned to face left. All images are trained and tested at a resolution of $128 \times 128$.

## A3. Ablation on Memory Bank Vectors and Keypoint Numbers

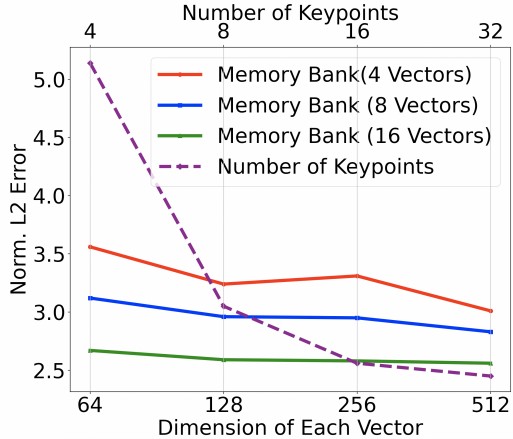

*Figure A1.* **Ablation of our PPL method on memory bank vectors and number of keypoints in Human3.6m.** The upper horizontal axis is the number of keypoints (ranging from 4 to 32) and the lower horizontal axis is the dimension of memory bank vectors (ranging from 64 to 512). The dashed purple line is for ablations on number of keypoints and the solid lines are for ablations on memory bank vectors.

## A4. PPL Results of Keypoint Detection on Occluded Images

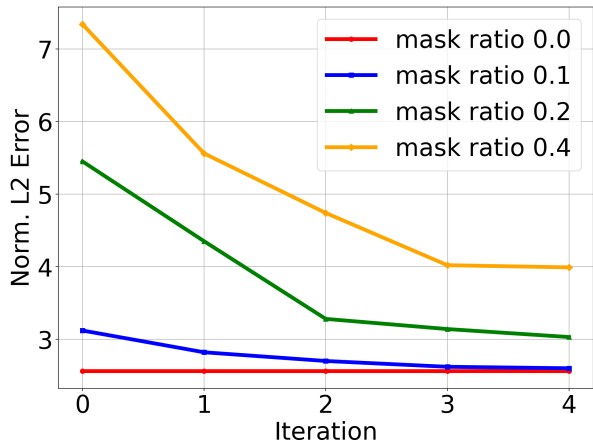
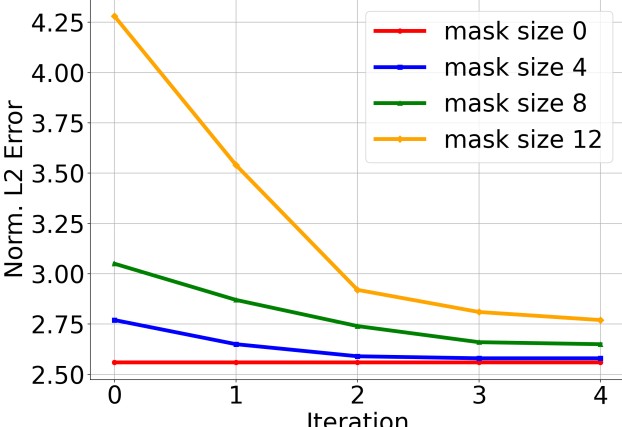

*Figure A2.* **PPL results of keypoint detection as a function of number of inference iterations on images with RandomMasking from Human3.6m.** The "mask ratio" in the legend specifies the masked proportion on the $32 \times 32 = 1024$ image patches.

*Figure A3.* **PPL results of keypoint detection as a function of number of inference iterations on images with CenterMasking from Human3.6m.** The "mask size" in the legend refers to the width and height of the masked region, on 1024 image patches.

## A5. Retrieval and Distillation of the proposed Hierarchical Memory in PPL

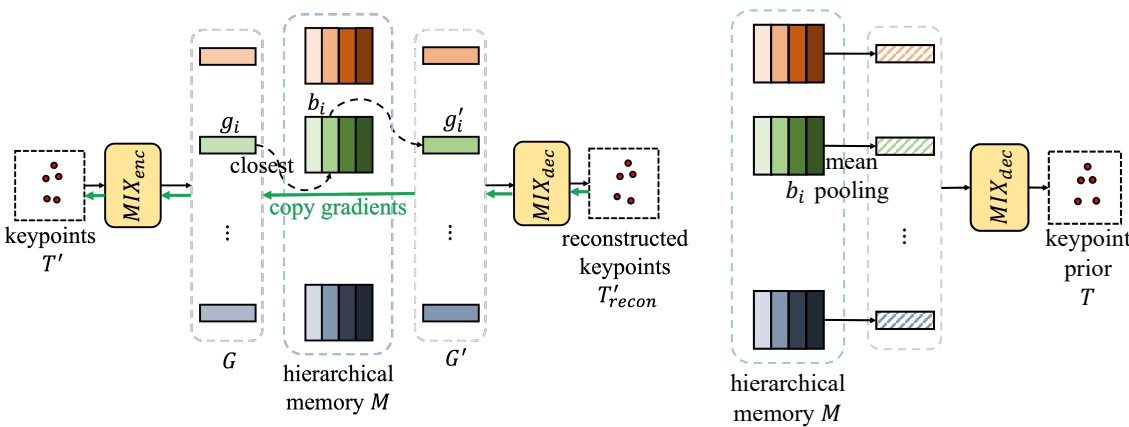

(a) Keypoint configuration reconstruction.

(b) Memory distillation.

*Figure A4.* **Retrieval and distillation of the proposed hierarchical memory in our PPL.** (a) The hierarchical memory $M$ is trained to reconstruct the keypoints $T'_{recon}$. $T'$ is encoded into $m$ tokens by the MLP-Mixer blocks $MIX_{enc}$. Each token $g_i$ retrieves its closest vector $g'_i$ in memory bank $b_i$. The resulting $m$ vectors are decoded by the MLP-Mixer $MIX_{dec}$ into the reconstructed keypoints $T'_{recon}$. The green arrows indicate the gradient flows during backpropagation based on the reconstruction of keypoint configurations. See **Section 3.4** for training details. (b) The hierarchical memory $M$ is distilled into the keypoint prior $T$. Vectors in every memory bank $b_i$ are mean-pooled into one vector, and the resulting $m$ vectors are decoded by $MIX_{dec}$ into the keypoint prior $T$. See **Section 3.3** for details.

## A6. Training Techniques

To ensure convergence and stability during the training of PPL, we introduce three gradient dettachment techniques: (1) To address the broken gradient issue during the quantization step from $G$ to $G'$, we adopt the approach from VQ-VAE (Van Den Oord et al., 2017). Specifically, our PPL copies the gradients of $G'$ to $G$ for backward propagation, allowing the gradients to flow through the quantization step. (2) The hierarchical memory $M$ is updated using an exponential moving average to smooth the gradient updates, particularly during the early stages of training when $G$ can be quite noisy. This approach helps stabilize the learning process and ensures that $M$ retains more reliable information over time. (3) For $M$ to

learn effective representations of $G$, it requires an accurate estimation of $T'$, which depends on a good prior $V$ distilled from $M$. This creates a chicken-and-egg problem that complicates training. To address this, we introduce two gradient detachments to separate the training processes. First, we detach the gradients from $T$ and train the keypoint transformation and image reconstruction pathway, as shown by the red arrows in **Figure 2**. Second, we detach the gradients from $T'$ to train the memory encoder and decoder, $MIX_{enc}$ and $MIX_{dec}$, as indicated by the green arrows in **Figure A4(a)** in **Appendix A5**.

## A7. Additional Visualizations

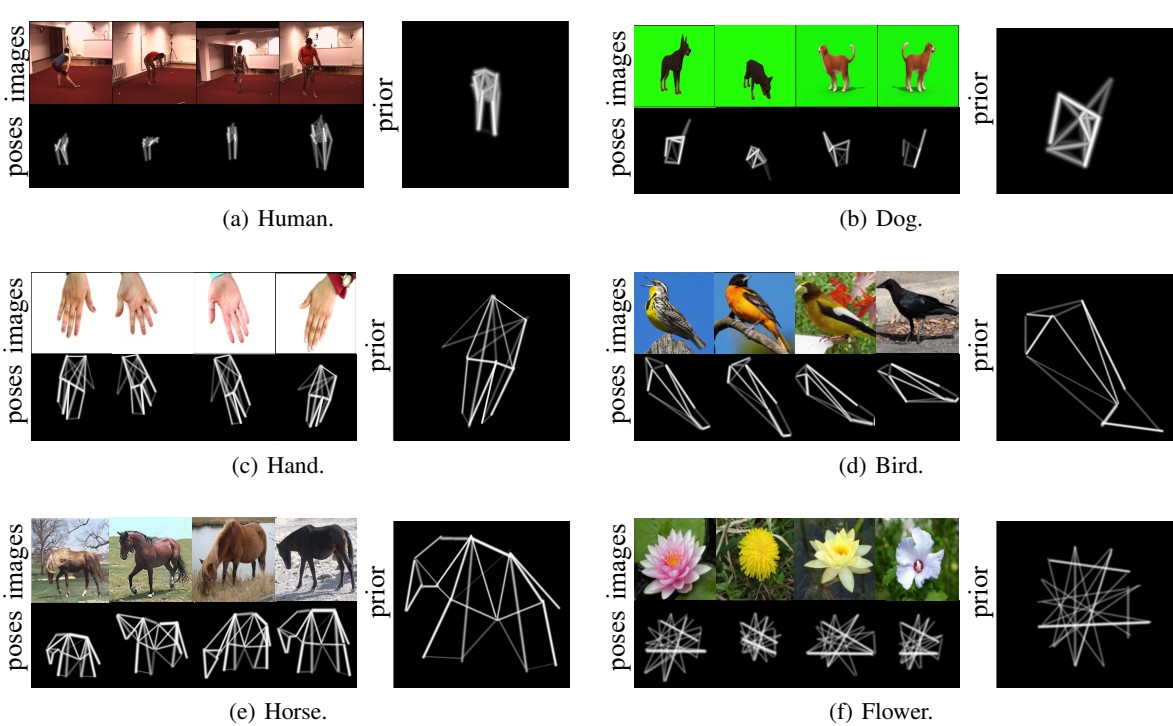

(a) Human.

(b) Dog.

(c) Hand.

(d) Bird.

(e) Horse.

(f) Flower.

*Figure A5.* **Additional Visualization on (a) Human3.6m, (b) Youtube dog videos, (c) 11k-Hands, (d) CUB-200-211, (e) Horse, and (f) Flowers.** Columns 1-4 of every subfigure show the original images and corresponding pose estimation results by PPL. The column 5 of every subfigure shows the learned prior for the category.

