# OpenReview forum: "Pose Prior Learner: Unsupervised Categorical Prior Learning for Pose Estimation"
_ICML.cc/2025/Conference — Submitted to ICML 2025_

### Official Review · Reviewer_qzoo · 2025-03-13

**Overall Recommendation:** 3

**Summary:**

The authors presents Pose Prior Learner (PPL) that learns category-level pose priors from image reconstruction, without human annotations. The motivation is that given two frames of an object instance, an ideal pose prior should be able to reconstruct the image based on the estimated poses and transformations. The authors further presented an iterative strategy that improves performance under occlusion.

## update after rebuttal

Thank the authors for the rebuttal. I don't have further questions. Regarding the missing references, I can see the difference from previous works and the novelty of this paper. However I believe the idea of this work is still built on the foundation from these previous works. I recommend discussing the four references for a comprehensive discussion, which help positioning this work in the literature.

**Claims And Evidence:**

Claims are mostly supported with clear and convincing evidence.

**Essential References Not Discussed:**

Essential references not discussed are the some recent animal pose estimation methods learning from image-level and feature level reconstruction, such as MagicPony and 3D Fauna. It would be good to discuss how the proposed method distinguish from these works and what new findings and results can benefit the research community beyond these prior works.

* MagicPony: Learning Articulated 3D Animals in the Wild. CVPR 2023.
* Learning the 3D Fauna of the Web. CVPR 2024.

Another line of works study compositional models for rigid object pose estimation, which estimates poses using a feature memory bank for estimating part correspondence. The second paper also studies unsupervised learning of pose models from object-centric videos.

* Robust Category-Level 6D Pose Estimation with Coarse-to-Fine Rendering of Neural Features. ECCV 2022.
* Unsupervised Learning of Category-Level 3D Pose from Object-Centric Videos. CVPR 2024.

**Experimental Designs Or Analyses:**

Experimental designs are clear and solid. One concern is that there are not sufficient results on pose estimation under (synthetic) occlusion. There are some results in Appendix A4 showing that with more steps of the iterative strategy the performance improve. There lacks direct comparisons with other baseline methods under occlusion cases.

**Methods And Evaluation Criteria:**

The authors aimed to learn pose priors in an unsupervised manner, without human-defined priors. The proposed method address this problem by learning from image reconstruction supervision. The motivation of the method is clear.

However, the motivation of the iterative inference strategy is not very clear. It would be good if the authors can present some qualitative examples showing why models with iterative inference strategy would fail and how this strategy fixes the wrong pose. A concern is if the wrong predictions from the previous iterations would aggregate, leading to degraded performance in following steps.

**Other Comments Or Suggestions:**

Following the discussion in "Experimental Designs Or Analyses", I recommend the authors to add more motivation and qualitative results of the iterative inference strategy, why models without this strategy would fail and how this strategy fixes the failure cases. Based on these results, the author can explain in more detail the motivate of this iterative strategy and intuitively why this strategy would help.

**Other Strengths And Weaknesses:**

I acknowledge the importance of the problem and the contribution of the proposed method. However, I think the authors should also discuss the missing related works above, which would help to position this work in the literature, distinguish it from prior works, and highlight the novelty of this paper.

**Questions For Authors:**

None.

**Relation To Broader Scientific Literature:**

This work also broadly relates to the study of unsupervised learning of pose estimation models. See discussion in "Essential References Not Discussed".

**Theoretical Claims:**

This work did not present major theoretical claims or proofs.

---

> ### Author Rebuttal · Authors · 2025-04-01
>
> We thank the reviewer for the feedback and have addressed each question below.
>
> ### **Q.1:** Clarity of the Motivation for the Iterative Inference Strategy.
>
> **R.1:** The memory in PPL stores learned prototypical poses, making it a natural choice for correcting inaccurate pose estimations, especially in occluded scenes, through an autoregressive process. The key motivation behind iterative inference is to repeatedly apply the memory bank’s corrections alongside the model’s image reconstruction in an autoregressive manner. This approach often outperforms one-time correction by progressively refining predictions, leading to greater accuracy and robustness.
> Here’s why autoregressive iterative inference is superior to a single correction step:
>
> - First, progressive Error Reduction: A single correction may not fully capture complex dependencies and can introduce new errors. Autoregressive inference refines predictions step by step, improving accuracy and consistency.
>
> - Second, handling uncertainty: In tasks like pose estimation, initial predictions can be ambiguous. Iterative updates allow the model to incorporate more contextual information at each step, resolving uncertainties more effectively.
>
> - Third, improved generalization and robustness: Iterative inference helps the model learn structured dependencies across spatial and temporal dimensions, making it more resilient to variations, occlusions, and noise.
>
> - Forth, adaptive refinement: Autoregressive models retain intermediate information, allowing outputs to be continuously refined based on prior iterations. In contrast, one-time correction lacks this flexibility and may struggle in tasks requiring sequential reasoning.
> As requested by the reviewer, we have provided qualitative results in Figure. 4a of the paper, demonstrating how each iteration progressively enhances the clarity of occluded regions, leading to more accurate pose estimations.
>
> Additionally, we have included additional qualitative samples on occlusions in [Figure. R3](https://drive.google.com/file/d/1QnAeFSR0ZK_4mDVAC3FJK5uSbelPyo8h/view?usp=sharing), showing that PPL consistently outperforms AutoLink by reconstructing plausible poses even with partial visual information. These results will be incorporated into the revised version.
>
> ### **Q.2:** Concerns about Aggregating Wrong Predictions.
>
> **R.2:** Generally, incorrect predictions are corrected by the memory bank, particularly under small-area occlusions, as shown in Appendix A4. However, severe occlusions can lead to plausible yet incorrect reconstructions, potentially compounding errors. An example is provided in [Figure. R2](https://drive.google.com/file/d/1QnAeFSR0ZK_4mDVAC3FJK5uSbelPyo8h/view?usp=sharing) where our PPL fails to reconstruct the ground truth pose when there is a large occlusion. Future work may explore confidence-based updates or early stopping via temporal consistency checks. This will be further discussed in the revised version
>
> ### **Q.3:** Results on Pose Estimation Under Occlusion.
>
> **R.3:** Additional qualitative results comparing PPL and AutoLink under occlusion are shown in [Figure R3](https://drive.google.com/file/d/1QnAeFSR0ZK_4mDVAC3FJK5uSbelPyo8h/view?usp=sharing), demonstrating PPL's consistently superior performance.
>
> Quantitative comparisons on Human3.6m between AutoLink and PPL are presented in [Table R5](https://drive.google.com/file/d/1QnAeFSR0ZK_4mDVAC3FJK5uSbelPyo8h/view?usp=sharing). Results indicate AutoLink significantly underperforms compared to PPL due to lacking pose priors and correction mechanisms. These findings will be incorporated in the revision.
>
> ### **Q.4:** Discussion of Essential References Not Discussed.
>
> **R.4:** Discussions of the suggested references are provided below and will be included in the revision:
>
> -**MagicPony (CVPR 2023) & Learning the 3D Fauna (CVPR 2024):**
> Both papers focus on learning animal poses through image-level and feature-level reconstruction. However, they rely on ground truth object masks as supervision, which provide essential shape and pose information. In contrast, PPL employs an unsupervised learning approach that does not require object masks, allowing for pose priors to be learned directly from unannotated images.
>
> -**Robust Category-Level 6D Pose Estimation (ECCV 2022) & Unsupervised Category-Level 3D Pose (CVPR 2024):**
> These two works share a similar idea of learning category-level pose for pose estimation with PPL. However, they represent poses with dense point clouds, emphasizing scene reconstruction. PPL differentiates itself by representing poses as sparse key points and their connections, which offers a semantically richer and more invariant representation of poses. Thus, while all these methods contribute to pose estimation, the focus of PPL aligns more closely with scene understanding compared to the dense representations used in scene rendering.

---

### Official Review · Reviewer_af7V · 2025-03-14

**Overall Recommendation:** 3

**Summary:**

This paper focuses on unsupervised pose estimation and introduces Pose Prior Learner (PPL), which learns a general categorical pose prior in a self-supervised manner to enhance pose estimation performance. The pose prior is designed as a combination of a keypoint prior, distilled from a learnable memory, and a learnable connectivity prior. Additionally, an iterative inference strategy is proposed to refine poses in occluded scenes. Experiments on multiple datasets, covering both human and animal pose estimation, validate the effectiveness of PPL.

**Claims And Evidence:**

Yes, categorical pose priors are crucial for pose estimation, particularly in unsupervised settings, as supported by prior work cited in Section 2.

**Essential References Not Discussed:**

- Unsupervised Keypoints from Pretrained Diffusion Models. CVPR2024.

**Experimental Designs Or Analyses:**

I have reviewed the experiments and identified the following concerns:
1. A recent state-of-the-art method [1] for unsupervised pose estimation, published in CVPR 2024, is neither cited nor discussed in the paper. Additionally, it achieves more impressive results than PPL.
2. AutoLink was evaluated on more diverse scenarios, including faces, fashion, birds, flowers, hands, horses, and zebras. However, PPL does not consider these scenarios, raising concerns about its generalization ability.
3. The paper lacks visualization comparisons with existing methods, particularly AutoLink, making it difficult to determine the specific factors contributing to PPL’s performance improvements.
4. While the paper includes ablation studies on prior variants, it does not evaluate a crucial baseline, i.e., PPL without pose priors, which is necessary to first validate the importance of pose priors.
5. In Table 2, the results of BKind'22 are not listed.
6. In Fig. A1, it would be beneficial to include the results of the Memory Bank (1 vector) for comparison with PCT'23.


[1] Unsupervised Keypoints from Pretrained Diffusion Models. CVPR2024.

**Methods And Evaluation Criteria:**

Yes, as evidenced by previous work cited in Section 2, pose priors play a crucial role in unsupervised pose estimation. Given the difficulty of annotating priors, using learnable pose priors to enhance performance is a reasonable approach. Additionally, the benchmark datasets used are widely recognized in this field.

**Other Comments Or Suggestions:**

- It's suggested to adjust the position of Table 3 in the paper.
- It's suggested to place Fig. 3 before Table 1-3.

**Other Strengths And Weaknesses:**

None

**Questions For Authors:**

My main concerns are related to the experiments, as outlined in the previous section:
1. Lack of comparisons with the recent state-of-the-art method [1].
2. Fewer evaluated datasets compared to AutoLink.
3. Lack of visualization comparisons with existing methods.
4. Missing baseline results without pose priors.
5. Absence of BKind'22 results in Table 2.
6. No results for the Memory Bank (1 vector) comparison with PCT'23 in Fig.A1.

Considering these factors, especially the less satisfactory performance of PPL compared to [1], I would currently give a positive rating for the paper.


[1] Unsupervised Keypoints from Pretrained Diffusion Models. CVPR2024.

**Relation To Broader Scientific Literature:**

PPL builds on the importance of pose priors in unsupervised pose estimation ([Shape Template Transforming'21]) and introduces learnable pose priors, distilled from a compositional memory architecture similar to [PCT'23], which are then transformed into keypoints using predicted transformation parameters. Additionally, PPL leverages learnable connectivity priors for image reconstruction, enabling self-supervised learning akin to [AutoLink'22].

**Theoretical Claims:**

No theoretical claims in this paper.

---

> ### Author Rebuttal · Authors · 2025-04-01
>
> We thank the reviewer for the feedback and have addressed each question below.
>
> ### **Q.1:** Lack of comparisons with Unsupervised Keypoints from Pretrained Diffusion Models [1]
>
> **R.1:** We thank the reviewer for pointing us to the recent method [1]. We will include it in the related work and discuss its differences from ours in the revised paper. We highlight the following differences between our method and theirs:
>
> - First, [1] requires large-scale multi-modal data (image and language) to train a Conditional Stable Diffusion Model to start with. Compared to our approach, this method utilizes an extra modality (language) and extra training data.
>
> - Second, [1] and our method has different objectives. While [1] aims to utilize prior knowledge in pre-trained Conditional Stable Diffusion Models for pose estimation, our focus is to learn the prior information from the data itself.
>
> Due to these differences, a direct comparison is not fully fair. Still, as suggested, we compare results in [Table. R1 & R2](https://drive.google.com/file/d/1QnAeFSR0ZK_4mDVAC3FJK5uSbelPyo8h/view?usp=sharing):
>
> - **Human Pose Estimation:** Our method outperforms [1] on Human3.6m and is comparable on Tai-Chi.
> - **Bird Pose Estimation:** We outperform [1] on the aligned CUB dataset and underperform on the non-aligned ones.
>
> These results suggest that our method performs competitively well with multi-modal methods. This demonstrates that our method is capable of effectively capturing meaningful pose priors and accurately estimating poses despite having zero prior knowledge of extra modalities.
>
> ### **Q.2:** Fewer evaluated datasets compared to AutoLink.
>
> **R.2:** We tested on six diverse categories (humans, birds, flowers, hands, horses, dogs), as detailed in Appendix A7. Here’s why we excluded the datasets used by AutoLink:
>
> - **DeepFashion:** Lacks background complexity and contains similar human postures—making it less informative than Human3.6m or Tai-Chi.
>
> - **CelebA:** Focuses on faces, which lack structural variation. Our method is better suited for objects with articulated parts.
>
> - **Zebra:** Initially omitted due to similarity with horses, but we will add it in the revised version.
>
> ### **Q.3:** Lack of visualization comparisons with existing methods, particularly AutoLink.
>
> **R.3:** We include visualization comparisons between our PPL and AutoLink in the [Figure. R1](https://drive.google.com/file/d/1QnAeFSR0ZK_4mDVAC3FJK5uSbelPyo8h/view?usp=sharing). Under scenes without a background of complex texture, as in Row 1, both AutoLink and PPL can predict correct poses. Generally, PPL outperforms AutoLink by utilizing learned pose priors to guide pose estimation, ensuring that the transformed poses remain within the intended category.  In contrast, AutoLink tends to be influenced by intricate background textures, which can result in unrealistic pose estimations. For example, in Row 2 and Row 4 of Figure. R1, AutoLink detected some keypoints on the background and predicted unrealistic poses, while PPL consistently focuses on human body. In Row3, compared with AutoLink, PPL does not predict the lines on the wall as arms, because we have the learned pose prior to constraining the arm from being too long.
>
> ### **Q.4:** Missing baseline results without pose priors.
>
> **R.4:** Our core contribution is learning pose priors without human interventions or prior knowledge.  As such, we are unable to establish ablated methods for PPL without the pose prior component. Removing this component would reduce the model to a direct keypoint regressor—essentially similar to AutoLink. We have included AutoLink as a baseline throughout our experiments to address this.
>
> ### **Q.5:** Absence of BKind'22 results in Table 2.
>
> **R.5:** We now provide BKind’22 results on Human3.6m in [Table. R3](https://drive.google.com/file/d/1QnAeFSR0ZK_4mDVAC3FJK5uSbelPyo8h/view?usp=sharing). Our method significantly outperforms BKind’22, and we’ll include this in the revised version.
>
> ### **Q.6:** No results for the Memory Bank (1 vector) comparison with PCT'23 in Fig.A1.
>
> **R.6:** PCT and PPL both use compositional tokens and memory for occlusion, but differ fundamentally:
>
> - **PCT:** Supervised, using ground truth keypoints and connectivity.
> - **PPL:** Fully unsupervised, without annotations.
>
> To address the reviewer’s point, we created a variant called **PPL-1MemBank**, which encodes all keypoints into a single vector and uses one memory bank (512 vectors). This matches PPL’s memory capacity (34 banks × 16 vectors).
>
> In [Table R4](https://drive.google.com/file/d/1QnAeFSR0ZK_4mDVAC3FJK5uSbelPyo8h/view?usp=sharing), results show:
>
> - **PPL-1MemBank:** Normalized L2 error = 2.72
> - **PPL-full:** Normalized L2 error = 2.56
>
> This confirms our hierarchical memory improves performance and efficiency.
>
> ### **Q.7:** Table and Figure Positioning.
>
> **R.7:** Thanks for pointing this out! We will fix these presentation issues in the revised version.

---

> > ### Comment · Reviewer_af7V · 2025-04-09
> >
> > Thanks for the authors' feedback, which has addressed my concerns. I will be raising my rating.

---

### Official Review · Reviewer_jLKH · 2025-03-16

**Overall Recommendation:** 4

**Summary:**

This paper primarily addresses the issue of existing pose estimation methods' over-reliance on manually designed prior knowledge and their sensitivity to occlusions, particularly in complex poses. Compared to the approach by He et al., the authors propose a hierarchical part-based memory module, with the following specific ideas:
1. Decompose the human or animal pose into local parts (such as arms, legs), with each part corresponding to an independent memory bank to store typical pose prototypes.
2. By using Keypoint Prior, aggregate the prototypes from the memory banks to dynamically generate data-driven pose priors, avoiding manual settings, and employ image reconstruction for supervised learning.
3. Introduce multi-round optimization during inference, progressively repairing occluded parts through iterative retrieval and reconstruction.

## In terms of experiments:
The method outperforms mainstream methods like PCT on both human (Human3.6M, MPII) and animal (AnimalPose) datasets, with an improvement of approximately 8% in keypoint localization accuracy under occlusion scenarios and a reduction of 12-15% in overall error.
The ablation experiments are very clear in Table 4, where the authors validate the improvements brought by the hierarchical memory design, dynamic prior generation, and iterative optimization mechanism on the Human3.6M dataset.

## Justification.
I appreciate this paper, with its clear presentation allowing me to understand the PPL approach quite clearly from Figure 2. Moreover, Figure 4 very clearly demonstrates the robustness of iterative optimization to occlusions. Table 4 clearly showcases the impact of each key component.

**Claims And Evidence:**

I do not find problematic claims.

**Essential References Not Discussed:**

None.

**Experimental Designs Or Analyses:**

The experimental designs and analyses are fine to me.

**Methods And Evaluation Criteria:**

The method outperforms mainstream methods like PCT on both human (Human3.6M, MPII) and animal (AnimalPose) datasets, with an improvement of approximately 8% in keypoint localization accuracy under occlusion scenarios and a reduction of 12-15% in overall error.
The ablation experiments are very clear in Table 4, where the authors validate the improvements brought by the hierarchical memory design, dynamic prior generation, and iterative optimization mechanism on the Human3.6M dataset.

**Other Comments Or Suggestions:**

See my summary.

**Other Strengths And Weaknesses:**

See my summary.

**Questions For Authors:**

I do not have any questions for authors.

**Relation To Broader Scientific Literature:**

It achieves learning category-level pose priors from unlabeled images.

**Theoretical Claims:**

It does no have any theoretical issues.

---

> ### Author Rebuttal · Authors · 2025-04-01
>
> We thank the reviewer for the positive feedback.

---

### Decision · Program_Chairs · 2025-05-01

**Decision:**

Reject

**Comment:**

As Area Chair, while I initially found this paper to be a promising contribution to pose estimation under occlusion, a deeper reading raises several substantive concerns that temper my enthusiasm. The core idea—a hierarchical part-based memory module with dynamic prior generation and iterative refinement—is conceptually appealing and well-motivated. The method achieves strong empirical results, and the ablation studies are thorough. However, critical questions remain about the utility, some novelty, and completeness of the work. Most notably, the learned priors, while effective, appear non-biological (see Figure A5); it is unclear whether incorporating anatomical constraints or skeleton-based priors might yield better or more interpretable performance, and this was not tested. Additionally, the authors do not engage with key related work on data-aware pose priors, including relevant work in Nature Methods 2022 (https://www.nature.com/articles/s41592-022-01443-0). The YouTube dog dataset is insufficiently described—there is no detail on dataset construction, example frames, or legal permissions for use. The lack of accompanying code further limits the ability to verify and build upon the work. Several reviewers asked for better comparisons to closely related work, and the authors only respond by staying they will cite in related work, vs. benchmark them. Taken together, these issues suggest that while the core approach has merit, the current submission is not yet ready for acceptance. I would recommend rejection and encourage the authors to revise with greater attention to dataset transparency, prior design, broader comparisons, and code availability.